# Clinical Phenotypes of Progressive Supranuclear Palsy—The Differences in Interleukin Patterns

**DOI:** 10.3390/ijms242015135

**Published:** 2023-10-13

**Authors:** Natalia Madetko-Alster, Dagmara Otto-Ślusarczyk, Alicja Wiercińska-Drapało, Dariusz Koziorowski, Stanisław Szlufik, Joanna Samborska-Ćwik, Marta Struga, Andrzej Friedman, Piotr Alster

**Affiliations:** 1Department of Neurology, Medical University of Warsaw, Kondratowicza 8, 03-242 Warsaw, Poland; dariusz.koziorowski@wum.edu.pl (D.K.); stanislaw.szlufik@wum.edu.pl (S.S.); josamborska@gmail.com (J.S.-Ć.); andrzej.friedman@wum.edu.pl (A.F.); piotr.alster@wum.edu.pl (P.A.); 2Department of Biochemistry, Medical University of Warsaw, Banacha 1, 02-097 Warsaw, Poland; dagmara.otto@wum.edu.pl (D.O.-Ś.); marta.struga@wum.edu.pl (M.S.); 3Department of Infectious and Tropical Diseases and Hepatology, Medical University of Warsaw, Wolska 37, 01-201 Warsaw, Poland; alicja.wiercinska-drapalo@wum.edu.pl

**Keywords:** progressive supranuclear palsy, PSP-P, PSP-RS, neuroinflammation, atypical parkinsonian syndrome, clinical phenotype

## Abstract

Progressive supranuclear palsy (PSP) is an atypical parkinsonian syndrome based on tau pathology; its clinical phenotype differs, but PSP with Richardson’s syndrome (PSP-RS) and the PSP parkinsonism predominant (PSP-P) variant remain the two most common manifestations. Neuroinflammation is involved in the course of the disease and may cause neurodegeneration. However, an up-to-date cytokine profile has not been assessed in different PSP phenotypes. This study aimed to evaluate possible differences in neuroinflammatory patterns between the two most common PSP phenotypes. Serum and cerebrospinal fluid (CSF) concentrations of interleukin-1 beta (IL-1β) and IL-6 were analyzed using enzyme-linked immunosorbent assay (ELISA) kits in 36 study participants—12 healthy controls and 24 patients with a clinical diagnosis of PSP-12 PSP-RS and 12 PSP-P. Disease duration among PSP patients ranged from three to six years. All participants underwent basic biochemical testing, and neutrophil-to-lymphocyte ratio (NLR) and platelet-to-lymphocyte ratio (PLR) values were calculated. Due to a lack of neuropathological examinations, as all patients remain alive, total tau levels were assessed in the CSF. Tau levels were significantly higher in the PSP-P and PSP-RS groups compared to the healthy controls. The lowest concentrations of serum and CSF interleukins were observed in PSP-RS patients, whereas PSP-P patients and healthy controls had significantly higher interleukin concentrations. Furthermore, there was a significant correlation between serum IL-6 levels and PLR in PSP-RS patients. The results indicate the existence of distinct neuroinflammatory patterns or a neuroprotective role of increased inflammatory activity, which could cause the differences between PSPS phenotypes and clinical course. The causality of the correlations described requires further studies to be confirmed.

## 1. Introduction

Progressive supranuclear palsy (PSP) is an atypical Parkinsonian syndrome neuropathologically defined as a tauopathy involving four microtubule-binding repeats (4R-tau). Definitive diagnosis requires post-mortem examination; however, current diagnostic criteria based on clinical manifestation [1] enable sensitive and reliable clinical diagnosis of probable PSP and possible PSP. Höglinger et al. [1] distinguished eight different phenotypes, although PSP with Richardson’s syndrome (PSP-RS) and PSP with predominant parkinsonism (PSP-P) are the most common, constituting 80–90% of all PSP cases [2]. Despite shared characteristic neuropathological features, PSP clinical manifestations are highly diversified, and the mechanisms underlying this phenomenon remain unclear. Among multiple theories aiming to at least partially explain the process of neurodegeneration, the inflammatory hypothesis appears to be one of the most documented.

Although it remains unclear whether neuroinflammation is the cause or rather an effect of pathological tau accumulation and neuronal loss, the presence of this process is definite. Initial reports date back to the nineties, with a paper describing increased glial activity in PSP patients [3]. The specific neuroinflammatory profile of PSP is not recognized as being among the factors associated with this process, of which interleukin-1 beta (IL-1β), IL-4, IL-6, and tumor necrosis factor-alpha (TNF-α) can be mentioned [4]. Indeed, IL-6 was associated with disease severity [5] in research based on the examination of 42 patients with probable PSP diagnosed using criteria released in 1996 [6]. The study [5] did not provide differentiation of the most common PSP phenotypes; moreover, no other interleukins except IL-6 were evaluated. Another study revealed a connection between IL-2 and peripheral immune responses in PSP [7], with increased levels of IL-2 in PSP linked with dysfunctional peripheral inflammation.

The neutrophil-to-lymphocyte ratio (NLR) and platelet-to-lymphocyte ratio (PLR) are non-specific indicators of systemic inflammation. Primary works on NLR involved the search for factors to assess intensive care unit patients. The ratio initially defined by Zahorec was associated with systemic inflammation and stress among critically ill patients [8]. Within the evolution of the NLR and the initiation of research concerning PLR, various studies on the role of NLR and PLR in predicting the course of other diseases, including vascular diseases, metabolic diseases, cancers, and neurodegenerative diseases, were introduced [9,10,11,12]. In the context of atypical parkinsonism, the evaluation of the ratios is based on a few works, though they show increased NLR values in PSP compared to Parkinson’s disease (PD) [13]. Studies by research groups analyzing the role of interleukins in PSP subtypes show that NLR is not a feasible tool for the differential evaluation of PSP and corticobasal syndrome [14]. Nevertheless, NLR increased significantly in those with PSP and PD compared to healthy volunteers, which suggests peripheral inflammation.

A study on synucleinopathic parkinsonisms revealed increased NLR values in PD and multiple system atrophy (MSA), though significantly higher PLR was only observed in PD [15]. This may lead to differences in peripheral inflammatory patterns between PD and MSA. The presented studies show that defining the role and significance of NLR and PLR in atypical parkinsonism is affected by significant obstacles, such as small numbers of patients in the studies, lack of neuropathological confirmation, and overlapping clinical manifestations.

The concentrations of IL-1 and IL-6 are not commonly evaluated in PSP; however, both factors were associated with the pathogenesis of atypical parkinsonism, including PSP and MSA [4]. In a study analyzing cerebral spinal fluid (CSF) concentrations of different interleukins and cytokines in patients with PD and atypical parkinsonian syndromes, it was revealed that PSP and MSA populations had significantly higher levels of C-reactive protein (CRP), TNF-α, IL-1β, IL-4, and IL-6 than the PD and control groups [16]. Furthermore, patients with MSA and PSP had similar cytokine profiles, while PSP patients had the highest concentrations of cytokines (statistically non-significant) [16]. However, the paper did not divide the PSP group based on clinical phenotypes. Another study revealed that serum IL-6 level correlates with disease severity in PSP, though it did not identify the phenotypes analyzed [5].

Increased inflammatory biomarkers have been found in the CSF of patients with MSA and PD, especially with PD dementia, and higher cytokine levels correlated with worse motor function and cognitive impairment [17]. Patients with PSP were also included in the study; however, this group formed the minority of the analyzed population (14/246), and the paper did not provide data on the PSP phenotype. 

Certain cytokines were evaluated in other tauopathies, mainly Alzheimer’s disease (AD) [18,19], with both factors linked to acute cognitive dysfunction, and IL-6 found to cause deviations in hippocampal gamma rhythm in mouse models [20]. An increase in IL-1β and TNF-α was a feature linked with periodontitis that initiated inflammatory processes leading to AD [21]. Increased levels of IL-6 were also detected in AD [22], and IL-6 trans-signaling was linked with cognitive deterioration in AD [23]. Another study showed a negative correlation between serum IL-6 levels and the volumetric measures within the Braak region, whereas no such correlation was observed between IL-6 levels in the CSF and possible atrophic changes in this region [24]. Interestingly, IL-1 was over-expressed in AD brains [25,26], with the mechanism interpreted as leading to plaque formation and an increase in acetylcholinesterase [25]. Additionally, the impact of inheriting IL-1A genetic polymorphisms was indicated as a feature related to an increased risk of AD at an earlier age [27]. Meanwhile, IL-1β was evaluated in other tauopathies, such as frontotemporal dementia (FTD) [28], though a study evaluating IL-1β levels in the CSF found no significant differences between FTD and controls.

As mentioned above, many studies have evaluated the possible role of inflammation in the pathogenesis of PSP and other neurodegenerative diseases. However, none of those studies investigate inflammatory-based causes of the clinical phenotype diversity observed among PSP patients. This study aimed to investigate whether PSP-RS and PSP-P differ in the context of CSF and serum cytokine profiles, which could explain differences in the clinical picture. Additionally, the study assessed the possible usefulness of non-specific inflammatory indicators, such as NLR and PLR, for evaluating cytokine expression in PSP. To the best of our knowledge, this is the first study exploring inflammatory-based differences in PSP phenotypes, which may clarify the origin of PSP clinical course complexity.

## 2. Results

### 2.1. Cerebrospinal Levels of Total Tau

Due to the lack of neuropathological confirmation of the diagnosis, as all study participants remain alive, total tau levels were evaluated in CSF. Higher levels were observed in the PSP-P and PSP-RS groups. Statistical analysis revealed significant differences between the PSP-P and healthy control groups and between the PSP-RS and healthy control groups. No statistically significant difference was found between the PSP-P and PSP-RS groups (Figure 1).

### 2.2. Serum Interleukins

#### 2.2.1. IL-1β

In the PSP subgroups, the serum IL-1β concentration was highest among patients with PSP-P. Statistical analysis revealed that the differences in IL-1β were significant when comparing the PSP-P group with the healthy control group (*p* < 0.0001) and the PSP-P group with the PSP-RS group (*p* = 0.0071). No statistically significant differences were observed between PSP-RS patients and the control group (Figure 2A).

#### 2.2.2. IL-6

The highest concentration of serum IL-6 was observed in PSP-P patients. A statistically significant difference in the IL-6 level was found between the PSP-P and the healthy control groups (*p* = 0.0222). No statistically significant differences were observed when comparing PSP-P to PSP-RS or PSP-RS to the healthy control group (Figure 2B).

### 2.3. Interleukins in the Cerebrospinal Fluid

#### 2.3.1. IL-1β

Statistical analysis revealed that the mean concentration of IL-1β in CSF was significantly higher in patients with clinical manifestation of PSP-P compared to patients with PSP-RS (*p* = 0.0088) and in the control group compared to the PSP-RS group (*p* < 0.0001) (Figure 3A). There were no statistically significant differences between the control group and patients with PSP-P. The highest concentration of IL-1β was observed in the healthy control group.

#### 2.3.2. IL-6

The analyzed data indicated that the highest concentration of IL-6 in CSF was observed among PSP-P patients. Differences in IL-6 concentrations were statistically significant between the PSP-P and PSP-RS groups (*p* = 0.0099) and between the PSP-P and healthy control groups (*p* < 0.0001). No statistically significant difference was observed between the PSP-RS and control groups. The data are summarized in Figure 3B, and the concentrations of interleukins among analyzed populations are summarized in Table 1.

Due to the age difference between the healthy controls and the PSP subgroups, statistical analysis was performed to assess the possible correlation between serum or CSF interleukin concentrations and age. No such correlations were found.

### 2.4. Correlations between Serum and Cerebrospinal Fluid Expression of Interleukins

No statistically significant correlations were found between levels of interleukins in the serum and CSF, as shown in Figure 4.

### 2.5. Neutrophil-to-Lymphocyte and Platelet-to-Lymphocyte Ratios—Results of Blood Analysis among Progressive Supranuclear Palsy Patients

Mean values of NLR, PRL, and blood analysis among PSP patients are presented in Table 2. Detailed data concerning blood samples are attached in the Appendix A.

### 2.6. Correlation of Interleukin Levels with Neutrophil-to-Lymphocyte Ratio and Platelet-to-Lymphocyte Ratio

The only statistically significant correlation between serum interleukin concentrations and NLR/PLR values was between serum IL-6 and PLR in the PSP-RS patients (*p* < 0.001; Pearson’s r = 0.7392). No statistically significant correlations were found between NLR/PLR and CSF interleukin levels (Figure 5 and Figure 6).

## 3. Discussion

Interpreting the differences in IL-1β and IL-6 levels revealed in the assessment of CSF and serum in PSP subtypes seems complex. The pace of clinical deterioration in PSP-RS and PSP-P may suggest that the neurodegenerative impact of the interleukins is present at different stages of the disease, occurring at an earlier stage in PSP-RS. This hypothesis is supported by the serum IL-1β and the serum and CSF IL-6 results, which may also explain why an increase in these factors occurred at an early stage in PSP-RS patients examined three to six years after disease initiation.

The main difference in the pathomechanism responsible for the distinct clinical phenotypes in PSP-P and PSP-RS may be based on the neuroinflammatory pattern. We hypothesize that in PSP-RS, characterized by rapid deterioration, the most intense inflammatory activity is present at the beginning of the disease, and it suddenly initiates further progressive neurodegeneration. In contrast, the inflammatory activity is rather constant in PSP-P, although less intense, and it, therefore, promotes less rapid neurodegeneration, which is reflected in a more favorable clinical course. This idea could explain the results obtained in the current study, as the disease duration varied between three to six years in all assessed patients. Indeed, the inflammatory activity should already have diminished at this point in PSP-RS patients, whereas it should remain in the shape of a more flattened curve in PSP-P and prolonged in time during the disease. Nonetheless, this hypothesis requires further investigation into changes in interleukin concentrations during different disease stages for distinct PSP phenotypes. To the best of our knowledge, this is the first study to analyze cytokine profiles in PSP variants, and there are no available data indicating possible changes in cytokine levels among PSP phenotypes during the course of the disease.

Another possible explanation that should be considered is the neuroprotective effect of the increased inflammatory activity. Indeed, increased inflammation could stunt the process of neurodegeneration as a protective mechanism, which could explain the higher concentration of cytokines in the PSP-P group compared to the PSP-RS group, and in the case of IL-1β in the CSF, the highest concentration was observed in healthy controls, which supports the idea of its beneficial impact. These findings are supported by the results of experiments conducted on mice, which demonstrate a neuroprotective effect of reactive microglia and IL-1β signaling on N-methyl-D-aspartate (NMDA)-induced retinal damage [29] and IL-6 signaling on repopulating microglia in traumatic brain injury [30]. Another paper confirmed the neuroprotective role of IL-6 in acute neuroinflammation, both in vitro and in vivo [31]. Some data indicate that the final result of IL-1β signaling could depend on the duration of exposure to this cytokine, promoting neuroprotection in the acute phase but causing neurodegeneration when exposure is prolonged [32]. Even though both theories are interesting, they are not fully supported by the results obtained. Therefore, this matter requires further research in order to be clarified.

The general lack of correlations between interleukin levels and PLR/NRL ratios could be explained by the low specificity of the ratios. The data suggest that PLR is more valuable in the context of neuroinflammation assessment in PSP; however, this finding requires further studies to be confirmed. The lack of correlations between serum and CSF levels of interleukins highlights the need to analyze both fluids separately, as blood sampling cannot replace CFS testing of neuroinflammatory activity evaluation.

The study was affected by certain limitations, including the relatively small sample size of 12 participants in each group. However, the limited number of patients was due to differentiating the two most common PSP subtypes. Previous studies on the inflammatory aspects of PSP did not distinguish the disease subtypes. Furthermore, the study was based on a single examination. Due to the pace of deterioration, especially in PSP-RS patients, undertaking a reliable follow-up was not achievable.

The study provides an overview of the differences in the levels of two parameters, IL-1 and IL-6; however, the goal of the study was to obtain an accessible and possibly simple method of examination, which could be feasible in clinical practice. The research was not based on a definitive diagnosis as all the patients included in the study are alive, and neuropathological examination was not possible.

To increase the validity of the results, the authors performed an additional evaluation of tau in the CSF, which showed similar levels in PSP-P and PSP-RS. This assessment was performed to reduce the risk of misdiagnosis of PSP-P/PD. Correlations between the serum and CSF levels of IL-1 and IL-6 in PSP and peripheral inflammatory factors have not been previously explored.

The study also stresses the issue insufficiently described in the contemporary literature concerning the differential diagnosis of PSP-RS and PSP-P [2]. However, over six years since the introduction of the most recent criteria of diagnosis, the differentiation of these two entities lacks efficient tools, which may be surprising regarding the fact that PSP-P, a more beneficial form of the disease, is associated with about 30% of the PSP cases. The boundaries connected with methods of examination are partly related to their possible implementation in clinical practice. Previously defined methods of assessment of PSP-P as Mesencephalon/Pons ratio, Magnetic Resonance Parkinsonism Index (MRPI) and its second version generally concentrated on the issue of differential diagnosis of PSP-P and PD [33,34] The early differentiation of PSP-RS and PSP-P remains an unexplored area. Certain perspectives showed a different tau distribution or discrepancies within the corpus callosum [35,36]. Nevertheless, the majority of works are based on a small number of patients. Some of the works suggest that although, at the early stage, PSP subtypes differ, their advanced stages at a certain moment begin to resemble one another [2].

## 4. Material and Methods

Samples analyzed in this study (blood and CSF) were obtained from patients with PSP-P or PSP-RS hospitalized in the Department of Neurology, Medical University of Warsaw. All diagnoses were based on current criteria [2], and none were confirmed by neuropathological examination as all patients remain alive. The population consisted of 12 patients with PSP-P and 12 patients with PSP-RS. The control group included a healthy population of 12 patients hospitalized in the Department of Infectious Diseases, Tropical Diseases and Hepatology, Medical University of Warsaw; only patients without a chronic neurological disorder were admitted because of the risk of neuroinfection; however, this diagnosis was excluded during the initial hospitalization. Characteristics of the population analyzed are presented in Table 3.

Blood samples (5 mL) were collected into test tubes without anticoagulant and centrifuged. CSF (10 mL) was collected by lumbar puncture. Serum and CSF were frozen at −80 °C until analysis. The IL-6, IL-1β, and tau protein levels were determined in the collected material, though tau was only measured in the CSF. All markers were measured using commercial enzyme-linked immunosorbent assays (ELISAs) (human (h) IL-6 HS and IL-1β ELISA kits from Diaclon SAS and a tau protein ELISA kit from Cloud-Clone Corp, Katy, TX, USA). Absorbance was determined at 450 nm using a plate reader. The concentrations of the tested markers were calculated based on standard curves.

All patients underwent basic laboratory testing, including blood morphology, CRP levels, and biochemical and electrolyte analysis. None of the patients had elevated markers of infection, such as CRP or leukocytosis. NLR and PLR were calculated by dividing the number of neutrophils by the number of lymphocytes and by dividing the number of platelets by the number of lymphocytes obtained from the same blood sample, respectively. The initial evaluation of the number of neutrophils and lymphocytes was performed using the Sysmex XT 4000i automated hematology analyzer in the Department of Laboratory Diagnostics of the Mazovian Brodno Hospital in Warsaw.

Statistical analysis employed GraphPad Prism 8 (GraphPad Software, San Diego, CA, USA) to calculate arithmetic means (X) with standard deviations (SDs). When assessing the statistical significance of differences between the means, *p* < 0.05 was assumed, and appropriate statistical tests were used. The Shapiro–Wilk W test determined the distribution (estimation of normality) of the analyzed variables, while the non-parametric Mann–Whitney U test compared the distributions of the variables in the two groups. Pearson’s correlation coefficients were used to assess the significance of correlations. One-way analysis of variance (ANOVA) (Kruskal–Wallis) followed by Dunne’s post hoc test analyzed the differences among the three groups.

## 5. Conclusions

The results obtained in this study indicate the existence of different inflammatory patterns among patients with distinct clinical phenotypes of PSP. Currently, the relevance of these findings is unclear, as the exact mechanisms underlying PSP phenotypes are unknown. However, the data provide insight into the pathological processes of tau-protein-based neurodegeneration.

A detailed exploration of the mechanisms of PSP pathogenesis could be used in clinical practice as a differential diagnosis method and for predicting the clinical course or expected lifespan assessment. Moreover, if the widely discussed hypothesis concerning the causative role of neuroinflammation in neurodegeneration could be confirmed, a thorough analysis of cytokine profiles seems crucial for future putative therapies aimed at altering the fatal course of the disease. Future therapies based on modifying the course of the disease could delay the clinical deterioration of the patients. The issue requires further studies; however, this is the first paper indicating the existence of different inflammatory processes among patients with various PSP clinical manifestations.

## Figures and Tables

**Figure 1 ijms-24-15135-f001:**
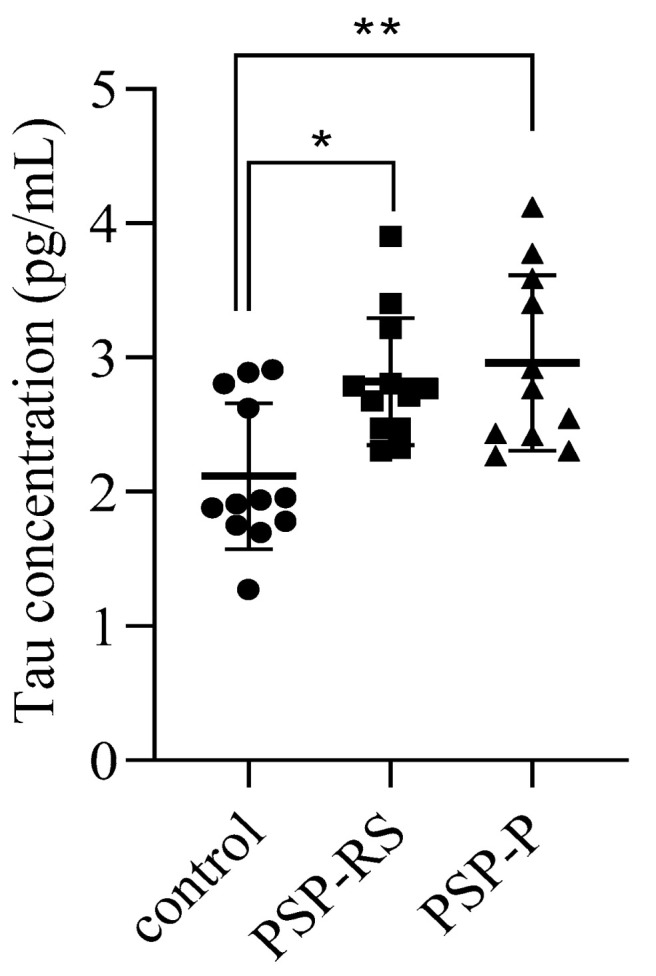
Levels of Tau in cerebrospinal fluid (CSF). Data from duplicate experiments are expressed as mean ± standard deviation. CSF tau levels were measured by enzyme-linked immunosorbent assay, and comparisons were made between the progressive supranuclear palsy with Richardson’s syndrome (PSP-RS) and PSP with predominant parkinsonism (PSP-P) groups and controls. Statistical significance was calculated using analysis of variance followed by Dunne’s post hoc test, * *p* < 0.05, ** *p* < 0.001.

**Figure 2 ijms-24-15135-f002:**
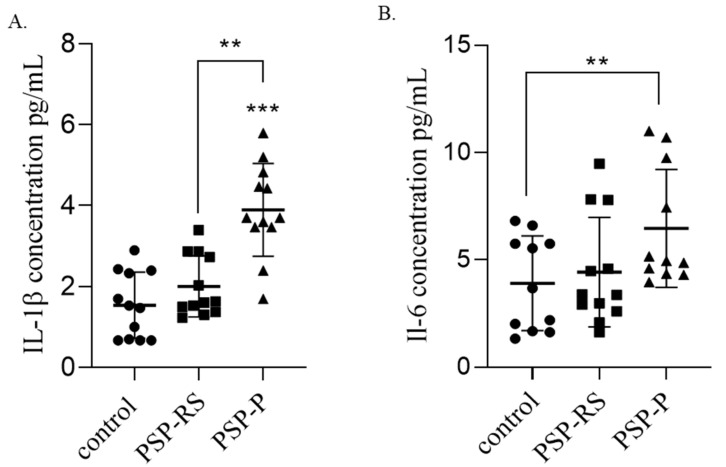
Comparison of serum interleukin-1 beta (IL-1β) and IL-6 concentrations among groups. Comparison of serum IL-1β (**A**) and IL-6 (**B**) concentrations between progressive supranuclear palsy with Richardson’s syndrome (PSP-RS) and PSP with predominant parkinsonism (PSP-P) groups and controls using enzyme-linked immunosorbent assays. Data of duplicate experiments are expressed as mean ± standard deviation. Statistical significance was calculated using analysis of variance followed by Dunne’s post hoc test, ** *p* < 0.001, *** *p* < 0.0001.

**Figure 3 ijms-24-15135-f003:**
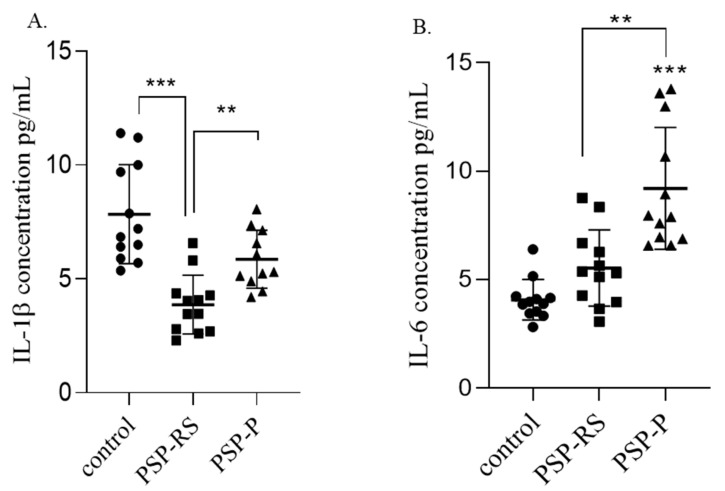
Comparison of interleukin-1 beta (IL-1β) and IL6 concentrations in cerebrospinal fluid (CSF) among groups. Comparison of CSF IL-1β (**A**) and IL-6 (**B**) concentration between the progressive supranuclear palsy with Richardson’s syndrome (PSP-RS), PSP with predominant parkinsonism (PSP-P), and control groups were measured by enzyme-linked immunosorbent assay. Data from duplicate experiments are expressed as mean ± standard deviation. Statistical significance was calculated using analysis of variance followed by Dunne’s post hoc test, ** *p* < 0.001, *** *p* < 0.0001.

**Figure 4 ijms-24-15135-f004:**
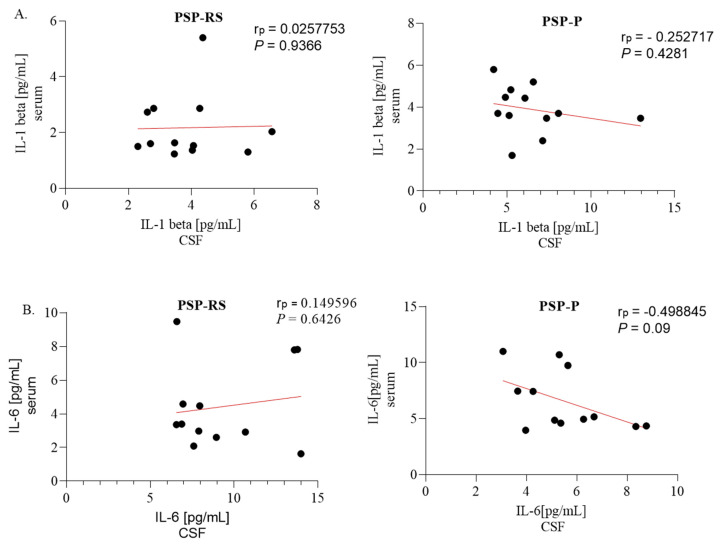
The association between serum and cerebrospinal fluid (CSF) levels of interleukin-1 beta (IL-1β) in progressive supranuclear palsy with Richardson’s syndrome (PSP-RS) and PSP with predominant parkinsonism (PSP-P) patients. (**A**) Serum IL-1β levels correlated with CSF IL-1 levels in PSP-RS and PSP-P, determined using Pearson’s correlation coefficient (rp). (**B**) Serum IL-1 6 levels correlated with CSF IL-6 levels in PSP-RS and PSP-P, determined using Pearson’s rp.

**Figure 5 ijms-24-15135-f005:**
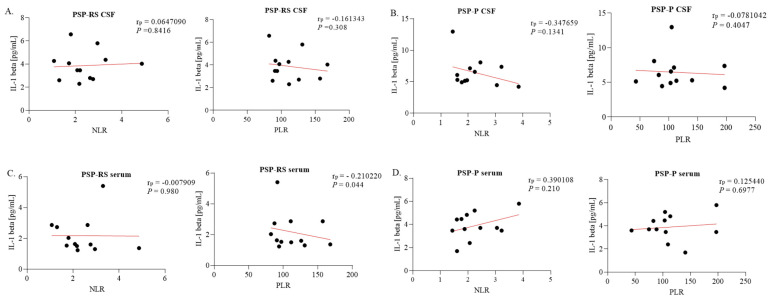
Correlations between interleukin-1 beta (IL-1β) levels and clinical characteristics. (**A**) IL-1β levels in cerebrospinal fluid (CSF) from progressive supranuclear palsy with Richardson’s syndrome (PSP-RS) patients, with neutrophil-to-lymphocyte ratio (NLR) and platelet-to-lymphocyte ratio (PLR) coefficients. (**B**) IL-1β levels in CSF from PSP with predominant parkinsonism (PSP-P) patients, with NLR and PLR coefficients. (**C**) IL-1β levels in serum from PSP-RS patients, with NLR and PLR coefficients. (**D**) IL-1β levels in serum from PSP-P patients, with NLR and PLR coefficients. The correlations were established by calculating Pearson’s correlation coefficient (rp).

**Figure 6 ijms-24-15135-f006:**
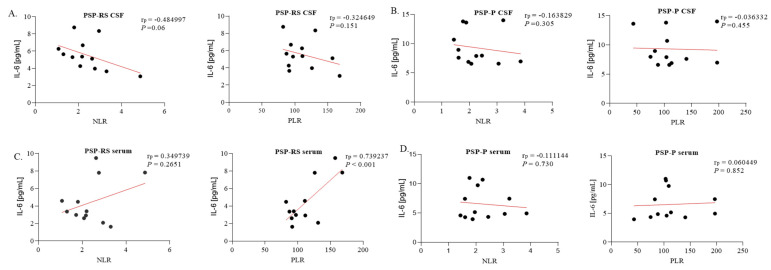
Correlations between interleukin-6 (IL-6) levels and clinical characteristics. (**A**) IL-6 levels in cerebrospinal fluid (CSF) from progressive supranuclear palsy with Richardson’s syndrome (PSP-RS) patients, with neutrophil-to-lymphocyte ratio (NLR) and platelet-to-lymphocyte ratio (PLR) coefficients. (**B**) IL-6 levels in CSF from PSP with predominant parkinsonism (PSP-P) patients, with NLR and PLR coefficients. (**C**) IL-6 levels in serum from PSP-RS patients, with NLR and PLR coefficients. (**D**) IL-6 levels in serum from PSP-P patients, with NLR and PLR coefficients. The correlations were established by calculating Pearson’s correlation coefficient (rp).

**Table 1 ijms-24-15135-t001:** Concentrations of interleukins among analyzed populations.

	Control Group	PSP-RS	PSP-P
**IL-1β in CSF (pg/mL)**			
mean value	7.838889	3.868889	6.448833
median	7.016667	3.75	5.683334
SD	2.17092	1.292941	2.384032
**IL-1β in the serum (pg/mL)**			
mean value	1.538889	2.172222	3.897222
median	1.5	1.616667	3.7
SD	0.816476	0.63	1.144637
**IL-6 in CSF (pg/mL)**			
mean value	4.071962	5.537339	9.286164
median	3.926681	5.331976	7.924135
SD	0.929435	1.762108	2.949349
**IL-6 in the serum (pg/mL)**			
mean value	3.910656	4.427359	7.182641
median	3.397149	3.68391	5.169043
SD	1.204799	1.510545	2.573167

**Table 2 ijms-24-15135-t002:** Blood sample analysis among progressive supranuclear palsy patients—mean values.

Diagnosis	Neutrophils (×10^3^)	Platelets (×10^3^)	Lymphocytes (×10^3^)	NLR	PLR
PSP-RS	4.43	210.17	1.96	2.41	112.29
PSP-P	4.65	232	2.20	2.26	113.23

**Table 3 ijms-24-15135-t003:** Characteristics of the analyzed population.

	PSP-P	PSP-RS	Healthy Control
Number of participants	12	12	12
Age (years) (mean)	55–80 (66.67)	64–75 (68.7)	35–69 (50)
Sex (male:female)	7:5	7:5	5:7
Disease duration (years) (mean)	3–6 (4.5)	3–6 (3.5)	-

## Data Availability

The datasets used and/or analyzed during the current study are available from the corresponding author upon reasonable request.

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
