# Peer review of "Clinical Phenotypes of Progressive Supranuclear Palsy—The Differences in Interleukin Patterns"

_ijms, 2023, doi:10.3390/ijms242015135_

Round 1

Reviewer 1 Report

The Authors evaluated evaluate possible differences in neuroinflammatory patterns between PSP-P and PSP-RS patients compared to healthy controls through concentrations of interleukin 1 beta and interleukin 6 in the serum and in cerebrospinal fluid. The manuscript is clear and well-structured. The cited references are useful and relevant. The experimental design is acceptable for gaining insights into neuroinflammatory mechanism but, as correctly highlighted by the Authors, the studied phenomena is complex and would require a longitudinal design in future. The sample included in the study was relatively limited as correctly evaluated by the Authors. The manuscript’s results could be reproducible based on the details given in the Methods. The figures and tables are appropriate. The conclusions are consistent with the studied topic but would require stronger future evidences as taken into account by the Authors. Conflict of interest and funding statements are adequate.

Here below, only one consideration:

1.      Line 115: Authors state “Tau protein level was determined only in the serum”. Please clarify this statement.

Author Response

Dear Reviewer 1, 

We are very grateful for your valuable comments.
Suggested change was implemented and confusing statement was clarified - tau protein level was determined only in the CSF – thank you for pointing that typo error. 

With best regards
Natalia Madetko-Alster

Reviewer 2 Report

Current study attempts to analyse inflammatory patterns of two types of progressive supranuclear palsy. The study is well designed. However, there are several issues that should be addressed before the publication.

1. In the introduction authors cite many studies, but they do not provide a summary that would highlight a knowledge gap that they aim to fill.

2. line 169 state that statistically significant difference was detected between control and PSP-P, but figure 2 shows the difference between PSP-RS and PSP-P.

3. The section 2.3 is difficult to understand and evaluate without a figure that would confirm the statement made in the section. Please, include the graph.

4. The title of the article and the conclusions presented in the abstract are too strong comparing to the actual results and the discussion that states that none of the mentioned theories can be confirmed. Conclusions sentence is very confusing, I would suggest to rephrase it in a more clear style. Also the word "explaining" is rather strong, authors are showing a series of correlations between type of PSP and concentration of certain ILs without proving the causality (which would be quite difficult).     

Some sentences are dense and a bit difficult to follow.

Author Response

Dear Reviewer 2, 

We are very grateful for your valuable comments. 
Please find below our responses. 

  1. In the introduction authors cite many studies, but they do not provide a summary that would highlight a knowledge gap that they aim to fill.

Ad 1. A paragraph: “As mentioned above, there were many studies concerning the possible role of inflammation in the pathogenesis of PSP or other neurodegenerative diseases. However, none of those studies investigate inflammatory-based causes of clinical phenotype diversity observed among PSP patients. The aim of this study was to investigate whether PSP-RS and PSP-P differ in the context of cytokine profile in cerebrospinal fluid (CSF) and serum, which could explain differences in the clinical picture. To the best of our knowledge, this is the first study exploring inflammatory based differences of PSP phenotypes which could possibly clarify the origin of PSP clinical course complexity” was added.

2. line 169 state that statistically significant difference was detected between control and PSP-P, but figure 2 shows the difference between PSP-RS and PSP-P.

Ad 2. The figure was corrected.

3. The section 2.3 is difficult to understand and evaluate without a figure that would confirm the statement made in the section. Please, include the graph.

Ad 3. Graphs were included.

4. The title of the article and the conclusions presented in the abstract are too strong comparing to the actual results and the discussion that states that none of the mentioned theories can be confirmed. Conclusions sentence is very confusing, I would suggest to rephrase it in a more clear style. Also the word "explaining" is rather strong, authors are showing a series of correlations between type of PSP and concentration of certain ILs without proving the causality (which would be quite difficult).  

Ad 4. The title and abstract were rephrased.

Some sentences are dense and a bit difficult to follow. 

The manuscript has been proofread by a native speaker of English who holds a PhD in Biomedical Science - certificate attached. 

With best regards
Natalia Madetko-Alster

Reviewer 3 Report

Dear Authors

It is a work in search of biomarkers in PSP-PD and as such has value. However, the sample is small, the markers analysed are few and sometimes not well presented, and a final message about the results is missing. 

Introduction:

line 52: (...) with a paper (...)

lines 56-58: I think you should rephrase this sentence. Was IL-6 evaluated in this study? reference (6). Note that this reference does not follow the style of the rest of the references.

A research based on the examination of 42 patients with probable PSP diagnosis no differentiation of the most common phenotypes of PSP was done and no other interleukins were evaluated. 

Line 71: The acronym of the tumor Necrosis Factor-alpha is described here but appears earlier in the text (line 55), and there is a list of abbreviations later.  Same with CSF in line 99. 

The introduction provides important information but should be a little more orderly. It is good that you want to give information on other tauopathies such as AD but it should be more concise, summarising the most important findings in terms of IL levels in a paragraph.

I recommend that all cytokine information on PD be concentrated; brain areas, CSF and plasma. Then mention the other thaupathies and at the end state the aim of the work.  

Material and Methods

Line 102: specimens maybe samples?

You have 3 groups so you cannot use t-test, I recommend using a non-parametric ANOVA since you have a small group of patients. 

The informed consent of the participants and the approval of the centre's ethics committee are missing.

Results

IL serum analysis should be the 2.1 section

Usually * is indicated as < 0.05 and ** is indicated as < 0.001

Figure 2A should indicate *** between controls and PSP-P

Please provide a summary of IL results to introduce table 2. 

In M&M section you indicate that Tau was studies in serum but now you present results from CSF.

Figure 3. Please indicate the differences between groups with lines. 

Please indicate the correlation of IL in their section and analyze if there was a correlation between serum and CSF expression for each protein, and tau and ILs. 

If tau concentration is measured as a possible indicator of taupathy (PD), it should be presented first. 

You should indicate the values of NLR and PLR, and what they are useful for. 

I should include in the main text or as supplementary material the results of blood analysis. 

Discussion

I do not get what is the final message you want to transmit. 

Looking forward to your comments. 

Best regards, 

The article can be read correctly but it wouldn't hurt to have a native speaker read it. 

Author Response

Dear Reviewer 3, 

We are very grateful fot your valuable comments that enabled us to improve the manuscript. Please find below our responses.

line 52: (...) with a paper (...)
Change implemented.

lines 56-58: I think you should rephrase this sentence. Was IL-6 evaluated in this study? reference (6). Note that this reference does not follow the style of the rest of the references.

A research based on the examination of 42 patients with probable PSP diagnosis no differentiation of the most common phenotypes of PSP was done and no other interleukins were evaluated. 

Reference [6] style was corrected, sentence was rephrased: “IL-6 was associated with disease severity [5] in research based on the examination of 42 patients with probable PSP diagnosis according to criteria released in 1996 [6]. The study [5] did not provide differentiation of the most common phenotypes of PSP; moreover, no other interleukins except IL-6 were evaluated.”

Line 71: The acronym of the tumor Necrosis Factor-alpha is described here but appears earlier in the text (line 55), and there is a list of abbreviations later.  Same with CSF in line 99. 

Abbreviations were corrected.

The introduction provides important information but should be a little more orderly. It is good that you want to give information on other tauopathies such as AD but it should be more concise, summarising the most important findings in terms of IL levels in a paragraph.

I recommend that all cytokine information on PD be concentrated; brain areas, CSF and plasma. Then mention the other thaupathies and at the end state the aim of the work.  

The introduction was rearranged. 

Line 102: specimens maybe samples?

Change implemented.

You have 3 groups so you cannot use t-test, I recommend using a non-parametric ANOVA since you have a small group of patients. 

T-test was not used, thank you for noticing typo error. Description of used tests was corrected.

The informed consent of the participants and the approval of the centre's ethics committee are missing.

Those data are presented in separate section – lines 379-382:
“Ethics approval and consent to participate: The studies were conducted in accordance with the local legislation and institutional requirements. The participants provided their written informed consent to participate in this study. The study was performed in accordance with the Declaration of Helsinki. Study was approved by Bioethical Committee of Medical University of Warsaw, KB/139/2020.”

Results

IL serum analysis should be the 2.1 section

Change implemented.

Usually * is indicated as < 0.05 and ** is indicated as < 0.001

Change implemented.

Figure 2A should indicate *** between controls and PSP-P

Corrected.

Please provide a summary of IL results to introduce table 2. 

Implemented.

In M&M section you indicate that Tau was studies in serum but now you present results from CSF.

Thank you for pointing the mistake, tau was assessed in CSF only – change was implemented.

Figure 3. Please indicate the differences between groups with lines. 

Change implemented.

Please indicate the correlation of IL in their section and analyze if there was a correlation between serum and CSF expression for each protein, and tau and ILs. 

Additional paragraph was added. 

If tau concentration is measured as a possible indicator of taupathy (PD), it should be presented first. 

Change implemented.

You should indicate the values of NLR and PLR, and what they are useful for. 

Implemented.  

I should include in the main text or as supplementary material the results of blood analysis. 

Blood analysis results were implemented in supplementary material.

Discussion

I do not get what is the final message you want to transmit. 

A separate paragraph (“conclusion”) was added to summarize the final message of the paper.

The article can be read correctly but it wouldn't hurt to have a native speaker read it. 

The manuscript has beed proofread by a native speaker of English who holds a PhD in Biomedical Science - certificate attached. 

With best regards
Natalia Madetko-Alster

Round 2

Reviewer 3 Report

Dear Authors

Thank you very much for your reply. 

I have no more comments.

Congratulations on your work. 

Best regards,